# Soccer, Sleep, Repeat: Effects of Training Characteristics on Sleep Quantity and Sleep Architecture

**DOI:** 10.3390/life13081679

**Published:** 2023-08-02

**Authors:** Patricia Frytz, Dominik P. J. Heib, Kerstin Hoedlmoser

**Affiliations:** 1Laboratory for Sleep, Cognition and Consciousness Research, Department of Psychology, University of Salzburg, 5020 Salzburg, Austria; 2Centre for Cognitive Neuroscience Salzburg (CCNS), University of Salzburg, 5020 Salzburg, Austria; 3Sport Psychology, Faculty of Sport Science, Leipzig University, 04109 Leipzig, Germany; 4Institut Proschlaf, 5020 Salzburg, Austria

**Keywords:** competitive sport, training intensity, sleep stage

## Abstract

Due to the high demands of competitive sports, the sleep architecture of adolescent athletes may be influenced by their regular training. To date, there is no clear evidence on how training characteristics (intensity, time of day, number of sessions) influence sleep quality and quantity. 53 male soccer players (*M* = 14.36 years, *SD* = 0.55) of Austrian U15 (*n* = 45) and U16 elite teams (*n* = 8) were tested on at least three consecutive days following their habitual training schedules. Participants completed daily sleep protocols (7 a.m., 8 p.m.) and questionnaires assessing sleep quality (PSQI), chronotype (D-MEQ), competition anxiety (WAI-T), and stress/recovery (RESTQ). Electrocardiography (ECG) and actigraphy devices measured sleep. Using sleep protocols and an ECG-based multi-resolution convolutional neural network (MCNN), we found that higher training intensity leads to more wake time, that later training causes longer sleep duration, and that one training session per day was most advantageous for sleep quality. In addition, somatic complaints assessed by the WAI-T negatively affected adolescent athletes’ sleep. Individual training loads and longer recovery times after late training sessions during the day should be considered in training schedules, especially for adolescent athletes. MCNN modeling based on ECG data seems promising for efficient sleep analysis in athletes.

## 1. Introduction

Along with numerous functions such as strengthening the immune system, memory consolidation, or regulation of the energy balance of the human body, our nocturnal sleep serves the physical and mental recovery from the exertions during the day. Nighttime sleep is divided into different stages [1]. The NREM sleep stages (=non-rapid eye movement sleep: light sleep, deep sleep; N1, N2, N3) and especially N3 sleep (Slow Wave Sleep; SWS, deep sleep) are particularly crucial for physical recovery, while in some studies rapid eye movement (REM) sleep has been linked to emotion regulation and motor learning (for a review, see [2]). Thus, both the first half of the nocturnal sleep period, when NREM3 sleep predominates, and the second half, when REM sleep is most pronounced, represent important phases for recovery. One of the factors used to determine sleep quality is the absolute amount of sleep one gets during the night. According to the American Academy of Sleep Medicine, the ideal amount of sleep for adolescents (14–17 years) is eight to ten hours [3]. Additionally, in their meta-analysis, Claudino et al. [4] indicated sleep efficiency (=actual sleep time expressed as a percentage of time in bed), sleep onset latency, wake episodes, and total wake episode duration as reliable markers to measure sleep quality in team athletes—defining it “as a variable of complex definition and diagnosis which depends directly on some parameters related to sleep architecture (=distribution of sleep stages; basic structural organization of sleep) such as sleep efficiency, latency and wakefulness duration”. Accordingly, another marker for healthy sleep is the appropriate distribution of sleep stages, with around 50% light (N1 and 2), 25% deep (N3) and 25% REM sleep throughout the night [5].

A special role is given to the sleep of competitive athletes. Due to the high physical and mental stress caused by intensive training sessions, competitions, and constantly high performance demands, the need for sufficient sleep is particularly high and, at the same time, very difficult to fulfill [6]. Studies show that adolescent athletes in particular only obtain seven to eight hours of sleep and that their subjective sleep needs are often not met [7,8,9]. Whitworth-Turner et al. [10] reported that youth elite soccer players aged between 18 and 20 years have a higher sleep latency, lower sleep efficiency, and a more irregular sleep–wake rhythm than non-athletes of the same age. The relationship between competitive sport and sleep is bidirectional [11]. Whereas insufficient sleep has a detrimental effect on sport performance [12,13], competitive sport also influences athletes’ sleep architecture (i.e., distribution of light, deep, and REM sleep). For this paper, we will focus on the latter effect. Daytime physical activity is generally known to have a sleep-promoting effect with more continuous deep sleep and longer sleep duration [14,15]. While some studies are indicating that trainings scheduled in the morning (i.e., >8 h before bed) [16] have a direct beneficial effect on the following night and that trainings during the evening (i.e., 10–12 h after wake-up time) decrease REM sleep and increase heart rate the following night [17], other studies did not find any time of day effects on sleep regarding the training schedule [18]. An important factor might be the type (aerobic, resistance, or interval) and regularity of the training [19,20]. Souissi and colleagues [21], for example, found that maximal aerobic exercise in the evening leads to more disturbed and fragmented sleep in trained subjects than afternoon exercise. Evidence of regular training in this field is very rare [22]. In their meta-analytic review, Kredlow and colleagues described that training duration positively moderated the beneficial exercise effects on sleep, with acute exercise showing a positive outcome on several sleep parameters like sleep architecture and sleep quantity, while regular training had a positive outcome on sleep onset latency. Conversely, increased training duration during a high-intensity training camp had a detrimental effect on the total sleep time of professional rugby players [23].

Despite the beneficial effects of daytime physical exercise, high-intensity training loads can have the opposite effect and lead to sleep deficits with longer periods of wakefulness and lower REM sleep continuity [15]. Previous studies investigating the influence of exercise intensity on sleep architecture have shown inconclusive results. Hrozanova et al. [11] discovered that mental strain and training load decreased REM sleep of junior endurance athletes during the following night. Short-term intensified training resulted in lower sleep efficiency in well-trained athletes in a study by Killer and colleagues [24]. Soccer players who maintained their regular training intensity during COVID-19 lockdowns showed lower subjective sleep quality than players with reduced training intensity [25]. On the other hand, Asian adolescent high-level athletes in high-intensity sports showed higher deep sleep, lower light sleep, and reduced wakefulness after sleep onset during regular training compared to athletes in low-intensity sports [26]. Knufinke and colleagues [27] could not find any effect of perceived training load on sleep quantity or sleep-stage distribution. Besides having sufficiently long sleep durations, participants showed relatively high sleep fragmentation. According to the authors, not only training intensity but other factors such as timing or number of training sessions should be considered as potential factors influencing subsequent sleep. Similar recommendations can be drawn from a recent review on the sleep of professional athletes [28]. Some of the included studies indicated a negative effect of physical activity on sleep, especially during certain phases of the season, while other studies found no effect. The extent of sleep degradation due to training should be assessed based on training styles, timing, frequency, and training load.

In the past, most studies on training effects on sleep included cohorts of moderately physically active people with low exercise stress or focused on exceptional situations (e.g., lockdown effects) or high-intensity phases of athletes (e.g., pre-season/ training camps, competitions, rehearsals) [23,24,25,28,29], but not on regular training in elite athletes. The present study aimed at investigating the influence of regular training on sleep in a precise way, i.e., analyzing the different characteristics like training load, time of day, and number of training sessions in one cohort of elite youth soccer players. We assessed sleep quantity and architecture using a novel approach of electrocardiography (ECG-)based nocturnal sleep analysis [30] and hypothesized that sleep quantity and quality are worse after days with perceived high training intensity in juvenile athletes who perform sports regularly at a high level. Given the outlined discrepancies in the literature, we here exploratively hypothesize that (1) a higher perceived training load and (2) a later timing of the last training before bedtime would lead to lower sleep quality. Conversely, we expected that (3) a longer total exercise duration during the day would cause better sleep quality. To have a holistic view of the athletes in this sample, we (4) further investigated whether there would be an impact of chronotype, self-reported sleep quality, competition anxiety, stress, and recovery on objective sleep measurements.

## 2. Materials and Methods

### 2.1. Sample

A total of 53 male soccer athletes (*M* = 14.36 years, *SD* = 0.55) from elite Austrian U15 (3 teams; *n* = 45) and U16 (one team; *n* = 8) soccer teams participated in the study. Subjects and parents gave their written informed consent before study inclusion. The study was performed in accordance with the Declaration of Helsinki and was approved by the local ethics committee. During the study, athletes followed their habitual sleep–wake and training schedules, including school attendance and test matches. Data collection per team lasted 14–42 days in total, depending on their training schedule (see Appendix A).

### 2.2. Study Design

Figure 1 graphically depicts the study process.

The study was conducted between 2018 and 2021 at a soccer academy (see also [35]). To give an overview, only the measurement methods and parameters relevant to our hypothesis presented in this manuscript are described below. Study protocols as well as the training and school schedules of athletes did not change due to the COVID-19 lockdown. At the beginning, the athletes filled out the German version of the Morningness–Eveningness Questionnaire (D-MEQ; adapted to adolescents) [31] and the Competition Anxiety Inventory-Trait (WAI-T) [32].

The D-MEQ [31] is a 19-item, self-rated questionnaire to assess the chronobiological preferences of participants. Values between 16 and 73 can be obtained. Low sum scores are associated with the eveningness type (16 to 36), intermediate scores point to the intermediate type (37 to 50), and high scores characterize the morningness type (51 to 73).

The WAI-T [32] is a self-rated questionnaire to measure sports competition anxiety. The global scale yields a score from 4 to 16, with higher scores indicating higher anxiety. Item scores range from 1 (“not at all”) to 4 (“very much”) and belong to one of three components. Somatic anxiety is defined as the physically perceivable component of anxiety that manifests itself in signs of anxiety such as palpitations, clammy hands, or a sinking feeling in the stomach. Worry refers to the tendency of athletes to develop self-doubt and specific worries before competitions or to form negative expectations. The third component, concentration deficits, is the tendency of individuals to be distracted by external influences, for example, spectator reactions during ongoing competition. Together with worry, it represents the cognitive component of competition anxiety.

Once a week, the athletes answered the German versions of the Pittsburgh Sleep Quality Index questionnaire (PSQI) [33] and the Recovery–Stress Questionnaire (RESTQ) [34].

The PSQI [33] is a 19-item, self-rated questionnaire to measure sleep quality and disturbance on a global scale of 0–21 for the last four weeks. For this study, we changed the evaluation period to the last seven days. The seven components (sleep duration, sleep disturbance, sleep latency, daytime dysfunction due to sleepiness, sleep efficiency, overall sleep quality, and sleep medication use) yield a score ranging from 0 (“no difficulties”) to 3 (“great difficulties”). Subjects with scores on the global scale between 0 and 5 are defined as good sleepers, whereas scores of 6–10 or >10 indicate significant sleep disturbances and chronic sleep disorders, respectively.

The 24-item short version of the German RESTQ [34] is a self-rated tool to assess seven stress scales and five recovery scales, which yield two separate scores: overall stress and overall recovery of the past week.

During the entire data collection, the athletes kept a daily sleep log on their mobile phones (based on [36]) in the morning right after waking up and in the evening right before going to sleep. The questionnaires were either administered in paper form by LimeSurvey (LimeSurvey GmbH, Hamburg, Germany) or via the Trayn app (Trayn Inc., Vienna, Austria) (see Appendix A). After study participation, athletes received feedback on their individual sleep–wake behavior during a final assessment.

In the evening sleep log, they provided information about their daily activities, which included details on when and how long the athletes took a nap during the day. This included the question, “Did you participate in a training/match today?” [Yes/No] to categorize the respective days into (1) resting, (2) training, and (3) match days. Additionally, the subjective intensity of the training or match (“How intense was the training/match?”: [very easy—very hard]) had to be rated on a scale from 0 to 100. Further, the number of training sessions was assessed. Subjective training intensity was standardized using z-transformation as rating scales changed throughout the assessment between the different testing cohorts (2018–2021) for reasons of greater data accuracy. The timing of training sessions was derived from training schedules that were provided by the coaches. Each training session lasted about 90 min.

The morning log was used to assess subjective sleep, including sleep duration (“How long did you sleep?”), sleep latency (“How long did it take until you fell asleep?”), and sleep quality (“How well did you sleep?”: [very bad—very good]). The athletes also reported their morning fatigue (“I currently feel… [extremely overloaded, exhausted]—[very refreshed, freshly rested]”).

Sleep–wake activity of the athletes was assessed via MotionWatch 8 actigraphs (Cambridge Neurotechnology, Ltd. Actiwatch ©; Cambridge, UK) throughout the whole measurement (at least 14 days).

For three consecutive days, the athletes wore the ECG device eMotion FAROS 180° (Bittium Biosignals Ltd., Kuopio, Finland, 2018) to measure heart rate variability (HRV) and classify sleep stages. The “Faros” sensor, including three adhesive electrodes, was attached to the athletes’ chest. Epoch-by-epoch four-class sleep staging was performed on inter-beat interval (IBI) time series extracted from the ECG recordings using a multi-resolution convolutional neural network (MCNN) as implemented in the Nukkuaa™ sleep app (cf. Topalidis et al.) [30]. Sleep parameters were assessed using the hypnograms provided by the MCNN model (see Figure 2). The MCNN model used in this paper was developed in our group and is now used in a mobile app that aims to treat insomnia complaints (https://www.nukkuaa.com (accessed on 4 July 2023)). In Topalidis et al. [30], we could show that our MCNN, which was trained on almost 9000 full PSGs (full-night EEG, EMG, EOG and ECG recordings), is able to classify sleep into four classes (Wake, Light Sleep, Deep Sleep, and REM Sleep) with an accuracy of 81% (κ = 0.69) solely using inter-beat intervals as a source of information. In contrast, trained human experts agree in about 88% of cases when classifying sleep on the basis of PSG data. Hence, as human experts (the up-to-now gold standard in sleep classification) do not at all show near to perfect inter-rater reliability, contrasting the performance of the MCNN (81%) to the maximal accuracy one can expect given the stated agreement rates of human experts (88%), we claim that our MCNN reaches a relative agreement of 92% with respect to human experts.

Both the actigraphs and the ECG devices offer the possibility of setting markers for relevant events. The athletes set these at the beginning (“lights out”) and at the end (“lights on”) of the nights. All subsequent analyses were performed on a time-in-bed interval based on these markers. An overview of all sleep parameters derived from our MCNN models based on ECG measures can be found in Appendix B.

### 2.3. Statistics

For the analyses reported in this manuscript, only the days with ECG recordings were analyzed (see [35] for further analyses including actigraphy recordings). Only objective sleep parameters derived from ECG sensors via the MCNN model were used to determine sleep quantity and sleep stage distribution. Sleep stages were analyzed using the percentage of sleep stage duration in relation to total sleep time to prevent sleep stages with shorter or longer sleep durations from being under- or over-estimated. In addition, actigraphy-based sleep quantity parameters were analyzed for comparison between measurement techniques. Actigraphy evaluation was performed using MotionWare software version 1.1.20 (CamNtech Ltd.; Cambridge, UK) via the “Sleep Analysis” function.

We used the D-MEQ sum score to measure chronotype, one averaged PSQI sum score for the whole study period per athlete to indicate self-reported sleep quality, the WAI-T components somatic anxiety, worries, and concentration deficits to analyze competition anxiety, and the averaged RESTQ components stress and recovery over the whole study period, as well as the subscales fatigue and sleep quality, to measure stress and recovery of athletes. 

For analyses of the training characteristics, two/three data points per participant were compared based on the following conditions: we compared nights after (1) the most vs. least intense training day (two data points), (2) the earliest vs. latest training before bedtime (two data points), and after (3) resting days (no training), one training session, and two training sessions for each athlete (three data points). As athletes trained one or two times per day, we used the last training of the day to compare the earliest to last timing of training before bedtime. See Table 1 for the number of nights included for each condition.

Data were analyzed using IBM SPSS Statistics 28 (IBM Corp., Armonk, NY, USA, 2021). As parameters did not indicate a normal distribution, analyses were performed using Mann–Whitney–U tests and Spearman correlations. *p*-Values < 0.05 were considered statistically significant. Descriptive statistics for each participant are reported for ECG-based sleep quantity and sleep architecture measures. Raincloud plots (Figures 3–5) were plotted by https://www.bioinformatics.com.cn/en (accessed on 6 March 2023), a free online platform for data analysis and visualization.

## 3. Results

### 3.1. Descriptives

Table 2 provides an overview of the ECG-based sleep parameter measures over the monitoring period (means and standard deviations). On 28 days (13.4% of the study process), athletes indicated that they took a nap during the day with a maximum duration of 120 min and a minimum duration of 15 min (*M* = 50.89 min, *SD* = 23.06).

### 3.2. Questionnaires

For analyses of relationships between questionnaires and objective sleep parameters, we used the D-MEQ sum score indicating chronotype, the PSQI sum score indicating self-reported sleep quality, the WAI-T components somatic anxiety, worries, and concentration deficits indicating competition anxiety, and the RESTQ components stress and recovery, as well as the subscales fatigue and sleep quality indicating stress and recovery. Objective sleep parameters were derived from ECG devices. According to D-MEQ scores, the sample consisted of 9 evening (17.0%), 35 intermediate (66.0%) and 8 (15.1%) morning types. PSQI scores indicate a distribution of 91.5% good sleepers (*N* = 43; PSQI score < 5) and 8.5% bad sleepers (*N* = 4; PSQI score > 5). There was a statistical trend for a negative correlation between PSQI scores and the number of awakenings lasting longer than two minutes (NOA 2) (*r_s_* = −0.289, *p* = 0.051, *n* = 46).

The somatic anxiety component (*M* = 6.29, *SD* = 1.84) of the WAI-T showed statistical trends for longer wake times within the first to last sleep epoch (WTSP) (*r_s_* = 0.274, *p* = 0.052, *n* = 51), increased wake after sleep onset (WASO) (*r_s_* = 0.271, *p* = 0.054, *n* = 51), and a higher number of awakenings (NOA) (*r_s_* = 0.244, *p* = 0.085, *n* = 51); the subscale concentration deficits (*M* = 4.96, *SD* = 1.24) revealed a statistical trend for decreased light sleep (*r_s_* = −0.236, *p* = 0.096, *n* = 51). Descriptives and significant correlations of questionnaire scores with sleep parameters are shown in Table 3. Scatterplots of every significant correlation can be seen in Appendix C.

### 3.3. Perceived Training Intensity

For this analysis, we compared ECG-derived sleep parameters on nights after the least and most intense training for each athlete. Analyses showed a significantly higher wake time within the first to last sleep epoch (WTSP) (*U* = 368.000, *Z* = −2.264, *p* = 0.024), a trend for higher wake after sleep onset (WASO) (*U* = 397.000, *Z* = −1.892, *p* = 0.058), and a trend for a higher number of awakenings longer than two minutes (NOA 2) (*U* = 402.000, *Z* = −1.857, *p* = 0.063) for nights following the most intense training compared to the most moderate training day (Figure 3). Sleep quantity data (sleep duration, sleep onset latency) derived from actigraphy revealed no significant differences. Athletes took naps on 12.1% (*M* = 36.25 min, *SD* = 6.50) of the least intense training days and 12.1% (*M* = 50.00 min, *SD* = 12.25) of the most intense training days.

### 3.4. Timing of Last Physical Activity before Bedtime

For this analysis, we compared ECG-derived sleep parameters on nights after the earliest versus the latest training session before bedtime for each athlete. Participants showed a significantly longer TSP (duration from the first to the last sleep epoch) (*U* = 142.500, *Z* = −2.336, *p* = 0.020) and total duration of sleep within the TSP (TST) (*U* = 151.500, *Z* = −2.125, *p* = 0.034) during nights following the latest training session before bedtime compared to nights following the earliest training of the day (see Figure 4). The mean time of the earliest training session was 2:20 p.m. (*SD* = 2:49 h), and the mean time of the latest training session was 4:05 p.m. (*SD* = 0:07 h). There was no significant relationship between subjective training intensity and the timing of the last exercise of the day. Table 4 depicts the time of day distribution for each condition. Sleep quantity derived from actigraphy revealed no significant differences. Athletes took a nap on 9.1% (*M* = 60.00 min, *SD* = 0.00) of days with the earliest training and 36.4% (*M* = 57.5 min, *SD* = 32.79) of days with the latest training session.

### 3.5. Number of Training Sessions

For this analysis, we compared ECG-derived sleep parameters on nights after rest days versus one training session versus two training sessions for each athlete. Analyses revealed a statistical trend for higher wake time within the TSP (WTSP) (*U* = 387.000, *Z* = −1.679, *p* = 0.093) for nights following two training sessions compared to nights following one training session and no differences compared to nights following resting days. There was no significant relationship between subjective training intensity per session and the number of training sessions. Nights following one training session showed a significantly longer total duration of sleep within the TSP (TST) (*U* = 620.500, *Z* = −2.183, *p* = 0.029), higher REM (*U* = 583.000, *Z* = −2.524, *p* = 0.012), and lower NREM (*U* = 583.000, *Z* = −2.524, *p* = 0.012) than nights following resting days. Figure 5 displays the significant differences. For a more detailed representation, NREM is represented by deep and light sleep. Sleep quantity derived from actigraphy revealed no significant differences. Athletes took a nap on 23.3% (*M* = 43.89 min, *SD* = 12.42) of days with no training, on 7.5% (*M* = 50.00 min, *SD* = 14.14) of days with one training, and on 22.7% (*M* = 60.00 min, *SD* = 28.02) of days with two training sessions.

## 4. Discussion

The athletes showed an average sleep duration of 7:19 ± 0:40 h, which is below the recommended range for adolescents of eight to ten hours [3]. However, a duration of at least seven and a maximum of eleven hours is still evaluated as “appropriate” according to the American Academy of Sleep Medicine. With a mean light sleep percentage of 55.45 ± 5.54%, a mean deep sleep percentage of 25.56 ± 4.88%, and a mean REM sleep percentage of 18.99 ± 2.81%, the athletes showed a similar sleep architecture compared to non-athlete adolescents, as reported in the meta-analysis by Ohayon et al. [37] (i.e., 57% light sleep, 22% deep sleep, and 21% REM sleep for an average night of 8–10 h).

Our findings partly support the assumption that a subjectively higher training intensity leads to more disturbed sleep the following night. Athletes showed significantly longer wake times and a trend for more wake phases after days with training sessions subjectively rated as “most intense” compared to days with the least intense training session. These results are consistent with findings reporting disturbed sleep after high training intensities. According to a review by Stanley and colleagues [38], cardiac autonomic recovery based on HRV takes longer following high-intensity exercise compared to low-intensity exercise. As low HRV is linked to poor sleep quality in some studies [39,40], prolonged cardiovascular recovery might be an explanation for sleep disturbances after high-intensity training days. Former inconclusive findings might result from the use of either objective measurements (the definition of high-level sports) [26] or subjective measurements [11,27]. Our results indicate that it might be important to consider subjective ratings rather than objective training intensity scores (e.g., type of training plan, percentage of maximal oxygen consumption (%VO2Max) or maximal heart rate (%HRmax)), as it might give more insight to individual exhaustion levels. Nevertheless, to provide an all-encompassing analysis, future studies should collect both subjective and objective assessments, such as ratings of perceived exertion and %HRmax [41].

We found that athletes slept longer after their latest trainings compared to their earliest trainings, even though they indicated a nap during the day more often. These results might lead to the assumption that the time after later trainings during the day is too short for the athletes to fully recover, leading to a greater need for recovery during the night with longer sleep durations. On average, athletes went to bed earlier after late (22:36) than early (22:51) training. The findings support the restorative theory of sleep with a higher sleep need after later trainings to recover energy [42] and are in line with the results of a study by Edinborough et al. [43], who found that a delayed match start increased sleep duration, wake time, and time in bed. Nevertheless, recent studies that mostly focused on evening exercise concluded that late exercise during the day does not have a detrimental influence on sleep if it is performed at least 90 min (moderate-intensity exercise) [44] or 60 min before bedtime (vigorous exercise) [45,46] which is in line with our results as the latest training sessions ended at 5:45 p.m. Crucial processes that were mentioned are a sufficiently strong decline in endorphin levels and core body temperature before going to bed and the fact that the timing of exercise does not coincide with the increase in melatonin levels [47]. Whereas most studies on daytime effects focused on acute exercise effects, our findings contribute to closing the study gap addressed by Kredlow et al. [22], who pointed out the scarce evidence on time of day and regular activity effects in their meta-analytic review.

Our results revealed that the number of training sessions and thus the total duration of physical exercise during the day alter sleep quantity and architecture. In addition to a statistical trend close to significance for more wake time during the sleep period following two training sessions compared to one training session during the day, we observed a longer total sleep duration and more REM sleep following one training session compared to a rest day. This might lead to the conclusion that days with only one training session were most beneficial for sleep quality compared to the other days. The evidence that a longer acute exercise duration has a positive linear effect on sleep quality cannot be supported by our results, but this is in line with the finding that there is no general effect of a longer regular exercise duration on sleep parameters [22]. As one training session lasted around 90 min in our cohort, it can be assumed that rather a specific exercise duration might be most beneficial for sleep in athletes. A similar argumentation as for training intensity can possibly be used here: in principle, training has a positive effect on the sleep of young elite athletes, as long as the training load is not too high (too long or too intensive). Previous studies indicated an exercise duration of around one hour to enhance sleep quality [48]. Interestingly, athletes were the least likely to report taking a nap on days with one training session, whereas nap frequency was similar on resting days and days with two training sessions. Possible reasons are that athletes did not feel the need for daytime recovery on days with one training session or that athletes used the free time on resting days to catch up on sleep.

The outcome on mean sleep quality and state questionnaires indicated a positive relationship between the tendency towards an early chronotype and sleep quality, namely less wake time during sleep, less light and more deep sleep, and a higher sleep efficiency. In accordance with previous studies, this underlines the detrimental impact of a systematic misalignment of the sleep—wake schedule and the circadian rhythm of the biological clock on sleep quality—which is most pronounced in late chronotypes [49]. Our findings regarding the relationship between competitive somatic anxiety and sleep disturbances, like longer and more wake phases during the night and more light sleep, also align with previous studies. In our sample, having a higher state of competitive anxiety with somatic symptoms was related to worse sleep quality, even though the athletes did not have any important matches during the data collection phase. This finding indicates that somatic anxiety symptoms are not only relevant in competition phases and supports evidence regarding a relationship between sleep quality, anxiety, and somatic complaints [50]. In future studies, it might be of interest to investigate how athletes’ acute somatic complaints affect sleep quality and quantity, as they are at risk for pain and somatic complaints due to elite sport demands [51]. Athletes with higher stress scores in the RESTQ questionnaire revealed a higher need for recovery during the night with less wake time, lower light, and higher deep sleep, whereas recovery scores were positively related to higher wake times and light sleep as well as lower deep and REM sleep, pointing out a lower need for recovery during the night. As expected, higher fatigue in the RESTQ indicated lower subjective and objective sleep quality. Subjective sleep quality derived from sleep protocols supported these findings.

Sleep characteristics in this study were defined using ECG-based parameters of sleep quantity and sleep architecture. For comparison between the different objective measurement techniques, hypotheses were also tested with actigraphy-based sleep quantity parameters. No significant relationships between training characteristics and actigraphy-based sleep quantity were found, even if ECG-based sleep quantity revealed significant differences between parameters. This finding supports the notion that ECG-based sleep measurements using MCNN modeling using low-cost wearables are a promising alternative to established assessment tools like actigraphy in field research [30]. Wearables often use proprietary algorithms with little information regarding the specifics of their sleep detection. MCNN modeling might meet this problem [52]. Another benefit is the additional possibility of analyzing sleep stage distribution besides sleep quantity parameters, thus leading to a more precise sleep assessment.

### Limitations

The limitations of the present study primarily relate to the application of the measurement procedures in the field. Erroneous measurements cannot be excluded because the athletes wore the ECG and actigraphy devices independently at night without 24 h surveillance by the study directors. In addition to the application of the measurement devices, it should also be noted that the algorithms used for analysis are not immune to possible measurement errors. ECG-based neural network models were trained on 8898 full nights, and the analysis quality could be compared with expert inter-rater reliabilities, but the approach is still quite new and needs further validation [30]. The present study has attempted to contribute to this. Longer measurement periods and comparisons with polysomnography might address the above-mentioned issues. Furthermore, the study was susceptible to subjective assessments of training intensity by the athletes, which can always be biased by errors in self-assessment or by social desirability. Comparisons with objective training intensity using GPS or ECG measurements are recommended for future studies.

## 5. Conclusions

The adolescent athletes in this study showed a sleep duration in the lower range of the recommended duration with an adequate distribution of sleep stages at the same time. Our findings suggest that high subjective training intensity during the day is likely to disrupt sleep the following night and that late training times are associated with a higher need for sleep. The number of training sessions was relevant for sleep quality. One training session per day (approx. 90 min) was most beneficial for sleep quality. In general, our study could help fill the research gap regarding the interaction of different training characteristics [27] and regular training [22]. The strong association between higher somatic anxiety and lower sleep quality suggests that somatic complaints play an important role in the sleep of adolescent competitive athletes and should be considered in future studies.

As practical implications for coaches and managers, we recommend taking into account the individual subjective training intensity of athletes in training schedules and allowing sufficient recovery and sleep time after late training sessions. The MCNN model used to analyze sleep architecture seems promising for low-cost, time-saving analysis of sleep for competitive athletes in the field.

## Figures and Tables

**Figure 1 life-13-01679-f001:**
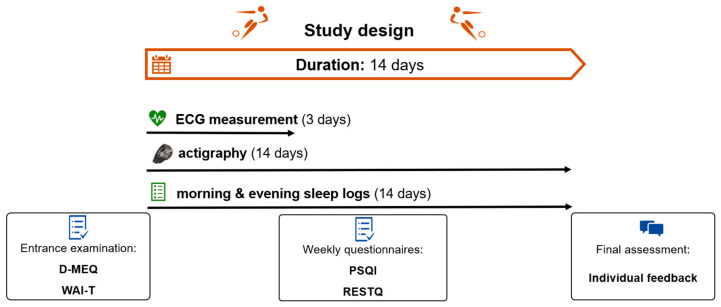
Study design. The study lasted a minimum of 14 days per team (see Appendix A for more information). It has to be noted that, for better understanding, we only depicted the minimum study duration of 14 days. Sleep logs and actigraphy were conducted throughout the whole study duration. Each ECG measurement lasted for three consecutive days. At the beginning, participants filled in the German version of the Morningness–Eveningness Questionnaire (D-MEQ) [31] and the Competition Anxiety Inventory-Trait (WAI-T) [32]. Once a week, they answered the Pittsburgh Sleep Quality Index (PSQI) [33] and the German version of the Recovery–Stress Questionnaire (RESTQ) [34]. After the study, athletes obtained individual feedback on their sleep–wake behavior during a final assessment session.

**Figure 2 life-13-01679-f002:**
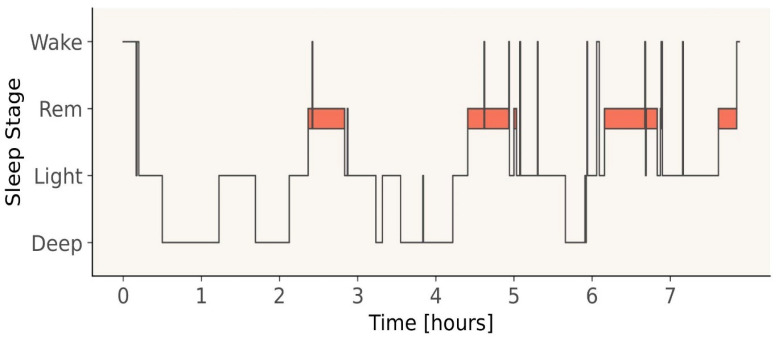
Example hypnogram based on the MCNN model. The hypnogram depicts one example night for one participant. Time [hours] indicates the sleep duration in the different sleep stages. Sleep stages (Wake, REM, Light, and Deep) were calculated by the MCNN model.

**Figure 3 life-13-01679-f003:**
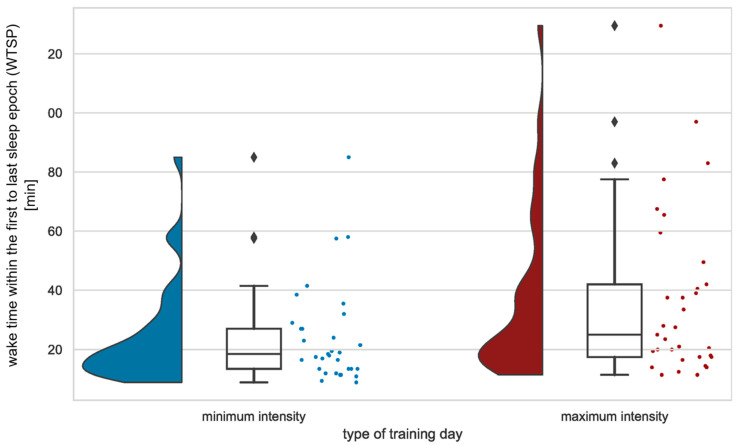
Raincloud plot of significant differences (*p* < 0.05) in wake time within the first to last sleep epoch (WTSP) between nights after minimum and maximum training intensity. Each box plot contains a horizontal line inside the box indicating the median, boundaries of the box indicating the 25th and 75th percentiles, and whiskers indicating minimum and maximum values. Scatter plots to the right and density plots to the left show the individual values and smoothed distribution of values. *N* = 33.

**Figure 4 life-13-01679-f004:**
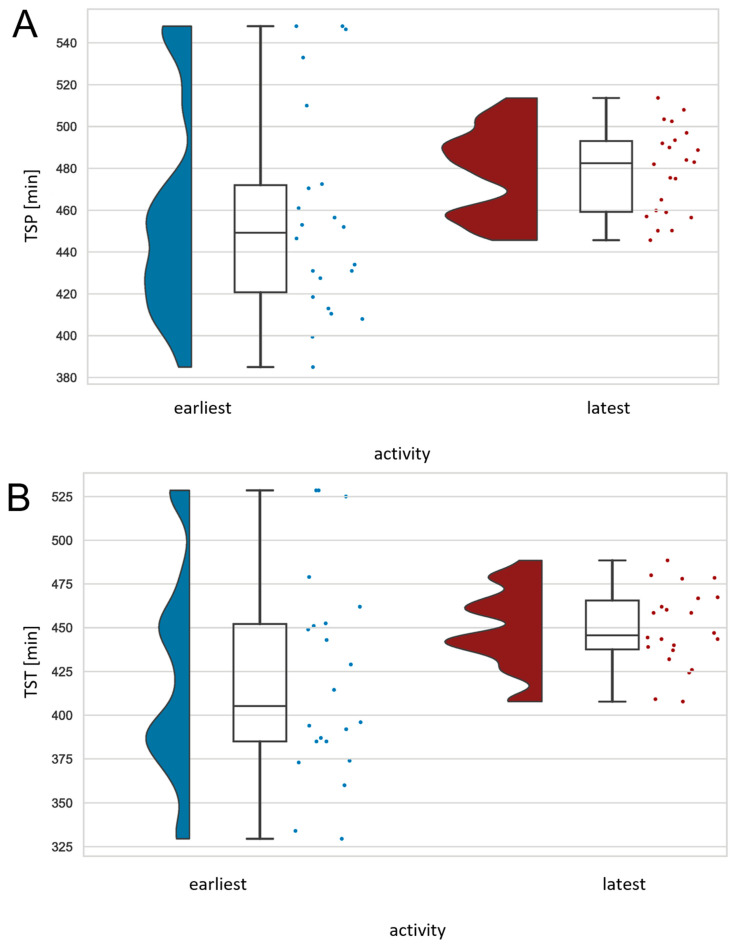
Raincloud plots of significant differences (*p* < 0.05) of TSP (**A**) and TST (**B**) between nights after minimum and maximum timing of last physical activity. Each box plot contains a horizontal line inside the box indicating the median, boundaries of the box indicating the 25th and 75th percentiles, and whiskers indicating minimum and maximum values. Scatter plots to the right and density plots to the left show the individual values and smoothed distribution of values. TSP = duration from the first to the last sleep epoch, TST = total duration of sleep within the TSP. *N* = 22.

**Figure 5 life-13-01679-f005:**
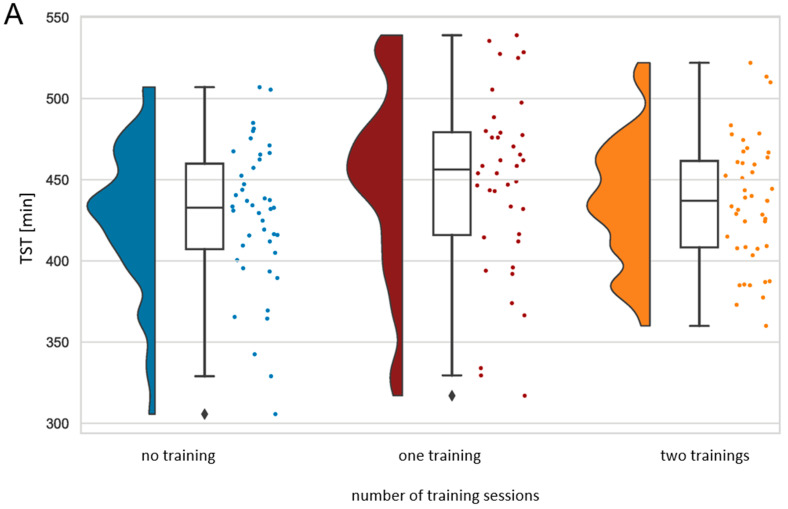
Raincloud plots of significant differences (*p* < 0.05) in TST (**A**) and sleep stages (**B**) between nights following no training, one training session, and two training sessions. For a more detailed representation, NREM is represented by deep and light sleep. Each box plot contains a horizontal line inside the box indicating the median, boundaries of the box indicating the 25th and 75th percentiles, and whiskers indicating minimum and maximum values. Scatter plots to the right and density plots to the left show the individual values and smoothed distribution of values. TST = total duration of sleep within the TSP. *N* = 42.

**Table 1 life-13-01679-t001:** Number of nights within and between each athlete included in each condition.

Training Characteristics	Nights within Each Athlete	Number of Athletes Included in the Analysis	Overall Count of Nights Included in the Analysis
Training intensity	2	33	66
Training time	2	22	44
Number of trainings per day	3	42	126

**Table 2 life-13-01679-t002:** Descriptive statistics for sleep quantity and sleep architecture measures derived from ECG devices.

		Mdn (IQR)
Sleep quantity	time in bed (min)	496.29 (472.38–519.72)
	total sleep time (min)	437.40 (414.25–458.58)
	sleep onset latency (min)	21.29 (12.78–32.54)
	wake after sleep onset (min)	41.00 (24.08–62.27)
	sleep efficiency (%)	87.57 (84.46–91.58)
Sleep architecture	percentage of light sleep (%)	54.97 (51.87–59.07)
	percentage of deep sleep (%)	25.52 (21.96–28.53)
	percentage of REM sleep (%)	18.80 (17.78–20.55)

Note. *N* = 53.

**Table 3 life-13-01679-t003:** Descriptives and significant correlations of questionnaire scores with sleep parameters derived from ECG devices.

Variable		M	SD	N	WTSP [min]	WAKE [min]	WASO [min]	LIGHT [%]	DEEP [%]	REM [%]	SE [%]	NOA 2	SOL 5 [min]	TST [min]	Subj. SQ	Morning Condition
D-MEQ	sum score	43.49	7.42	50		−0.416 **	−0.338 *	−0.317 *	0.366 **		0.375 **					
WAI-T	sum score	19.75	4.83	51												
	somatic anxiety	6.29	1.84			0.289 *		0.288 *				0.301 *				
	concentration deficits	4.96	1.24													
	worry	6.73	2.77													
RESTQ	stress	13.84	4.36	49	−0.391 **	−0.522 **	−0.432 **	−0.367 *	−0.337 *							
	fatigue	1.48	0.93			0.325 *					−0.345 *	0.341 *	0.353 *	0.353 *	−0.306 *	−0.306 *
	recovery	13.98	4.52		0.341 *	0.547 **	0.453 **	0.449 **	−0.313 *	−0.306 *						
	sleep quality	4.36	1.00												0.335 *	0.286 *

Note. WTSP = wake time within the first to last sleep epoch (TSP), WAKE = amount of wakefulness, WASO = wake after sleep onset, NOA 2 = number of awakenings within the TSP with a minimum duration of 2 continuous wake minutes, SOL 5 = time point after which the sleeper has spent at least five minutes (10 subsequent epochs) in a sleep stage other than wake (continuous sleep), TST = total duration of sleep within the TSP. * *p* < 0.05, ** *p* < 0.01.

**Table 4 life-13-01679-t004:** Distribution of timing for earliest and latest training before bedtime (as indicated in training schedules).

Condition	Timing	Overall Count of Nights Included in the Analysis
Earliest training before bedtime per athlete	7:30 a.m.	3
	3:00 p. m.	11
	4:00 p.m.	8
Latest training before bedtime per athlete	4:00 p.m.	14
	4:15 p.m.	8

Note. *N* = 22.

## Data Availability

The raw data supporting the conclusions of this article will be made available by the authors, without undue reservation, to any qualified researcher.

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
