# Peer review of "Soccer, Sleep, Repeat: Effects of Training Characteristics on Sleep Quantity and Sleep Architecture"

_life, 2023, doi:10.3390/life13081679_

Round 1

Reviewer 1 Report

Overall, the introduction provides a clear overview of the topic, highlights the importance of sleep for physical recovery and performance, and outlines the research objectives and hypotheses. However, it can be improved:

- What is the gap in scientific knowledge this study aims to contribute to clarify;

- There is a need to define some terms (NREM sleep, REM sleep; sleep efficiency and sleep architecture). Also, NREM do not appear in full lenght before in the article.

In the methods section, although it provides a detailed account of the study procedures, the part of data collection methods is hard to read. I think the use of shorter paragraphs could help making the understanding easier.

Overall, the article is well written, with some improvements needed mostly in the introduction and in the methods section.

Reviewer 2 Report

This manuscript presents an interesting field study that addresses the question of how regular training in high-intensity sports influences the sleep architecture of adolescent athletes. The objective was to examine the effects of different training characteristics, including subjective intensity, timing, and number of sessions, on different sleep parameters (e.g., sleep questionnaires) – whereas the focus is on heart rate based sleep analyses which provide data on different sleep stages (light, deep, REM). The study included 53 male soccer players from Austrian U15 and U16 elite teams. The study lasted 14-42 days per team, whereas ECG measurement was done for at least three consecutive days (nights?). The results in this manuscript included "only" the days with ECG recordings and should cover 159 nights. However, not all conditions were equally distributed, so the analysis was done with different n for each analysis. The findings revealed some interesting and statistically significant associations between training intensity, timing, and sleep duration, highlighting the importance of considering individual training loads and recovery time. Additionally, the study emphasizes the potential of neural networks for efficient sleep analysis based on ECG signals in athletes.

From my perspective, the manuscript's focus fits the scope of Life with the special issue on Research on Sleep Disorder and Sports.

Furthermore: I recommend publishing the manuscript but invite the authors to respond to the following minor points.

L103: "In accordance with literature, we hypothesized that (1) a higher perceived training load and (2) a later timing of the last training before bedtime would lead to more fragmented sleep, longer wake times and less REM sleep." As pointed out in the introduction the findings on training load on sleep parameters are much less clear (L85-96). Therefore, I recommend framing this study as explorative with rather open research questions. E.g., the conclusion from the cited study by Shapiro et al. (1981) is very optimistic.

L 172: In the evening sleep log, they provided information about their daily activities. This included the question "Did you participate in a training/match today?" [Yes/No] to categorize the respective days into (1) resting, (2) training and (3) match days. Additionally, the subjective intensity of the training or match ("How intense was the training/match?": [very easy - very hard]) had to be rated on a scale from 0 to 100. Further, the number of training sessions was assessed" --> At this point a Table would be helpful to clearly describe how many study nights from the potential 159 nights are included for each condition (separated in within and between descriptive data). Table 3 might point in this direction for the training time, but it is unclear if the count is within participants or between, e.g., 7:30 am with three could be one participants.

L217 "In addition, actigraphy based sleep quantity parameters were analyzed for comparison between measurement techniques." Maybe I missed but where can this analysis be found?

L221 "(1) most vs. least intense training day, (2) earliest vs. latest training before bedtime, (3) resting day vs. one training session vs. two training sessions)" again, the descriptive statistic of all conditions is not clearly reported.

L194 "Epoch-by-epoch four-class sleep staging was performed on inter-beat-interval (IBI) time series extracted from the ECG recordings using a multi-resolution convolutional neural network (MCNN) as implemented in the NukkuaaTM sleep app (cf. Topalidis et al.) [22]." The classification of the HR data is based on an algorithm previously published by the same research group. Therefore I would be more careful about the statement in L198, "Reliability tests of sleep staging accuracy of the MCNN attains levels comparable to expert inter-rater reliability.". Is this app open access? I could not find it. The authors should give more information on that MCNN because this app will attract most readers.

L212 and Table 2. Which data was statistically analyzed for the RESTQ and PSQI? The first or the weekly? Please describe more clearly.

Reviewer 3 Report

Dear authors,

I carefully read your manuscript aiming to fill some gaps in the literature about the effect of soccer training on sleep. Even though the specifical methodologies and different approaches to the problem, the reading is difficult, some important information is missing, and the statistical analysis does not lead to some conclusions exposed in the discussion section.

Moreover, I want to invite you to reflect on the chosen journal; maybe you can find a more appropriate MDPI journal focused on training and sport.

Please, find my comments below:

INTRODUCTION

1.       Especially in the second paragraph (lines 64-84), the citations refer to general physical activity, not training or elite athletes. You should focus on the literature of your topic and sample.

2.       In general, I suggest focusing more on your study and highlighting how your manuscript could fill some literature gaps.

3.       Citation 19 is more focused on the bidirectional relationship between physical activity and sleep: physical activity could influence sleep as well as sleep could influence physical activity or performance (see also https://doi.org/10.1016/j.physbeh.2022.113963, https://doi.org/10.3389/fphys.2023.1190956, https://doi.org/10.1007/s40279-014-0260-0, and https://doi.org/10.1249/JSR.0000000000000771).

4.       Lines 90-92, it is not clear the time frame.

5.       Line 99: which cohort?

6.       Line 106 is not clear.

7.       Line 109: what is general sleep? Maybe self-perceived or subjective sleep?

MATERIALS AND METHODS

8.       In general, this section needs to be better described since many details are missing and make the results section understand very hard.

9.       Line 119: “What does “on their time capacity” mean?

10.   Figure 1:

a.       Why is the sleep log from 7a.m. to 8p.m.?

b.       How many days did the participants wear the actigraph?

c.       At which time of the study protocol and how long did they wear the actigraph and the ECG device?

11.   Line 137: what did you do during the covid-19 lockdown? Could it have influenced your data?

12.   PSQI:

a.       you did not write that you administrated it also at the beginning of the study protocol

b.       it evaluated the prior 30 days, you cannot administer it every week; otherwise, there will be an overlap. You must have chosen a different questionnaire. These data are not reliable.

13.   Line 189, the wearing time is not specified.

14.   The actigraphy model is missing

15.   In this section, you do not speak about naps.

16.   I appreciate Appendix B; however, you must list there the sleep parameters used for the analysis; otherwise, it is very difficult to understand the results (please, see also https://doi.org/10.1080/07420528.2022.2157737 and https://www.camntech.com/Products/MotionWatch/The%20MotionWatch%20User%20Guide%201_2_28.pdf).

17.   Statistica analysis:

a.       It needs to be thoroughly reviewed since it does not follow the order and finding of the results

b.       You did not specify how questionnaires were analysed

c.       Lines 222-225 are not understandable

d.       Actigraphic software is missing

e.       If you were not able to analyse the evening session, how could you affirm a time-of-day effect?

f.        If data are not normally distributed, using median and quartile values rather than mean and standard deviation is better.

RESULTS

18.   The descriptive statistic of the sample is missing.

19.   In general, it is not clear from which device you obtained the sleep data. For each paragraph, it is better to specify to which data and source you refer (for example, table 1).

20.   Generally, it is unclear which parameters and groups you compared in each paragraph.

21.   Before exposing the correlation results, you need to introduce the parameters among which you performed the correlation analyses.

22.   Line 259, light?

23.   Line 290, p=0.058 significant?

24.   Line 294, what is X?

25.   Line 310, duration?

26.   You can speak about effects if you performed an ANOVA  or regression analysis; your analysis only highlights differences between different situations and correlations. Revised the manuscript or the analysis accordingly.

27.   Figure 3, which difference can the reader appreciate from the figure?

28.   Table 3 is superfluous and not understandable.

DISCUSSION

29.    Line 380, what is AASM?

30.   I cannot properly evaluate the discussion section due to the scarce readiness and comprehension of the results section. After the revision, I will evaluate the discussion when the manuscript becomes clearer. However, I suggest finding possible explanations for the results and not limiting the discussion to a comparison with previous studies.

English is good, but some sentences are wrong structured. 
